# Total Factor Productivity of Agricultural Firms in Vietnam and Its Relevant Determinants

**Mai Huong Giang [1,2], Tran Dang Xuan [1,*] [iD], Bui Huy Trung [2,3] and Mai Thanh Que [2]**

1   Graduate School for International Development and Cooperation (IDEC), Hiroshima University, Higashi Hiroshima 739-8529, Japan; giangmh.hvnh@gmail.com
2   Banking Academy, Dong Da District, Hanoi 1000004, Vietnam; trungbh.hvnh@gmail.com (B.H.T.); quemt@hvnh.edu.vn (M.T.Q.)
3   Graduate School of Social Sciences, Hiroshima University, Higashi Hiroshima 739-8525, Japan
*   Correspondence: tdxuan@hiroshima-u.ac.jp

**Abstract:** In Vietnam, agriculture is a key sector that promotes economic growth and poverty reduction. Therefore, improving productivity in agriculture is indispensable to the sustainability of the country. This research examined productivity and its determinants from 420 enterprises operating in agriculture. Productivity was measured as the total factor productivity (TFP) obtained from fixed and random effects models. The determinants of TFP including size and age, share of state and foreign ownership, export, accessibility to Internet and bank loan of firms, controlled for year fixed effects, were analyzed. It was shown that 74.6% companies in the agricultural sector were small in size (< 10 < 200 employees). Although the number of large firms (>300 employees) explained 10.6%, they had a remarkable and positive TFP (38.8%, $p < 0.01$), while both small and very small (<10, and <200 employees, respectively) had strikingly negative TFP values ($-71.3\%$ and $-32.1\%$, respectively, $p < 0.01$), as compared to the medium sizes (< 200 < 300 employees). It was also revealed that although foreign ownership was only 3.8% on average, it had a notably positive effect on TFP (55.0%, $p < 0.01$). In contrast, state ownership accounted for 30.7%, but it had a negative influence on TFP ($-7.5\%$). The export contributed a negligible and statistically significant effect to TFP (2.6%), which might be attributed to a limited number of firms (4.5%) having mobility in agricultural export. 73% received a bank loan, and only 18.2% had access to the Internet, but both of them yielded remarkable TFP values (18.5%, $p < 0.01$ and 3.4%, $p < 0.05$ respectively). The Hausman test indicated that the fixed effects (FE) model was more effective than the random effects (RE) model to estimate the TFP. The findings of this study suggested that reform efforts should focus on improving the productivity of small agricultural enterprises. In addition, foreign investment, effective use of bank loan and Internet accessibility should be further enhanced. The results of this study may provide insights for policymakers who aim to improve the productivity in agricultural enterprises and thereby contribute to the sustainable growth of the country.

**Keywords:** firm productivity; total factor productivity; agricultural sector; fixed effects; random effects

**JEL Classification:** D24; O13; Q12

## 1. Introduction

Productivity is defined as a ratio of output to a volume measure of input use (OECD 2001). This definition is consistent and has been accepted in a large number of studies (Syverson 2011). In recent years, productivity analyses using longitudinal micro-level data in various aspects have been

widely employed by economists and policy makers for both developed and developing countries. This is partly due to the increased availability of micro-level data and the development of different methodological approaches from the literature. Another major reason is that productivity growth at the micro level has been considered a key factor to yield economic growth at the macro level in the long run. To be more specific, an increase in productivity growth rate indicates that a higher output is produced with either the same or lower inputs. In other words, the input is utilized more efficiently due to the improvement of existing technology. This allows firms to reduce their costs and improve the quality of products and thus helps them to maintain or increase their competitiveness. Therefore, in order to draw more appropriate economic and industrial policies, it is important to have a complete understanding of productivity at the micro level.

This research aimed to explore the determinants of firms' productivity by employing a novel micro-dataset of 420 agricultural firms in Vietnam. There are several reasons for considering the Vietnamese agricultural sector as the ideal setting to conduct this study. First, to date, the empirical evidence on the determinants of firm productivity in emerging economies has been relatively limited. Most firm-level productivity studies were conducted on developed countries with richer sources of data. Second, over the past few decades, the agricultural sector in Vietnam has played a key role in poverty reduction and social stability (World Bank 2016a). However, despite of the remarkable outcomes achieved after the Doi Moi reform in the late 1980s and early 1990s, the contribution of the agricultural sector in Vietnam has not corresponded to its scale. Under the current period of industrialization and urbanization in Vietnam, the role of agriculture has even gradually become less important. The share of the agricultural sector in GDP was reduced from 40.5% in 1990 to 16.3% in 2016 (GSO 2016). Furthermore, agricultural productivity was still lower than that of industries and services. In 2016, average labor productivity in the agricultural sector was VND 32.9 million/labor while the industry, construction, and service sectors reached VND 112 million/labor and 103.5 million/labor, respectively (Mai and Yen 2018). Therefore, improving productivity is essential for the sustainable development of the agricultural sector. Third, despite the importance of the issue, this topic has not been widely researched for the case of the Vietnamese agricultural sector, with relatively little known concerning the factors that drive firm productivity. Most of the empirical studies have been focused on the use of different approaches to measure firm productivity in Vietnam. Using labor productivity and total factor productivity (TFP) measured by the index number approach, Doan et al. (2014) found that Vietnamese firm productivity increased significantly between 2000 and 2009. Huang and Zhang (2016) employed the Levinsohn and Petrin approach to measure TFP and showed that firms of various ownership structures had higher productivity after Vietnam joined the World Trade Organization in 2007. Tran (2014) examined both Levinsohn and Petrin, and Olley and Pakes approaches and observed that during the global financial crisis, the overall TFP of Vietnam declined from 2007 to 2008 but it increased in 2009. Concerning the determinants of TFP, Huang and Zhang (2016) examined the effect of ownership and trade on firm productivity in Vietnam. Doan et al. (2014) investigated the association between trade liberalization and the productivity growth of Vietnamese enterprises. Pham (2015) focused on the causality between export participation and firm productivity. Regarding the determinants of firm productivity in Vietnam as mentioned above, principally, they were examined by a single-variable framework. However, the multivariate approach, which is considered more comprehensive, has not yet been applied to investigate the productivity of the agricultural sector in Vietnam. This research was the first to explore productivity and its determinants from a data set of 420 agricultural enterprises from 2000 to 2009 by a multivariable approach. The findings of this study can be used for comparison across countries, especially for other developing economies. Furthermore, it allows us to answer some fundamental questions regarding the productivity performance of the Vietnamese agricultural sector in detail. This study also aimed at devoting proposals to improve the productivity of agricultural firms in order to contribute to the sustainable development of the country.

The remainder of this paper is organized as follows. Section 2 presents the methodology, empirical model, and data used in the study. In Section 3, descriptive statistics are discussed first and the main findings are then reported. Section 4 discusses the empirical results. Section 5 concludes the paper.

## 2. Methodology

### 2.1. Measurement of Firm Productivity

To measure productivity, there are two principal options including (i) partial factor productivity (PFP) and (ii) total factor productivity (TFP) (OECD 2001). The PFP is defined as a ratio between output and a specific factor input (capital or labor). Meanwhile, TFP relates to productivity counted by a ratio of output produced and an index of composite inputs. In other words, TFP is the weighted average capacity of all inputs (Owyong 2000). The output can be determined by gross output or value added. To measure TFP, there are two main directions in the literature: non-parametric approaches (TFP index and Data Envelopment Analysis (DEA)) and parametric approaches (Stochastic Frontier Analysis (SFA) and estimation of production function method). The frequent techniques to estimate the production function include the ordinary least square estimation, generalized method of moments, Olley and Pakes, and Levinsohn and Petrin approaches.

In this study, TFP was obtained from the production function to measure firm productivity. Accordingly, the index of relative TFP for each firm $i$ at time $t$ can be generally defined as follows:

$$\theta_{it} = \frac{Y_{it}}{f(K_{it}, L_{it})} \tag{1}$$

where:

$Y_{it}$: Output of firm $i$ at time $t$
$K_{it}$: Capital input of firm $i$ at time $t$
$L_{it}$: Labor input of firm $i$ at time $t$

$\theta_{it} = 1$ indicates the central tendency of TFP. If a firm's $\theta$ value is above 1, it indicated a high TFP relative to the other firms, whereas a value below 1 indicates low TFP. Rearranging (1) as an equation of $Y_{it}$, we have:

$$Y_{it} = f(K_{it}, L_{it})\theta_{it} \tag{2}$$

The next concern in (2) is the technology of production, which can be explained by different hypotheses. Among the others, the translogarithmic production and the Cobb–Douglas functions are the two most commonly used methods. It is argued that both approaches present good mathematic properties. However, the elasticity of the production to the inputs in the Cobb–Douglas is easier to interpret than in translogarithmic production. To be more specific, the translog technology generally suffers from a collinearity problem among the regressors (Kinda et al. 2011). Thus, in this study, we assumed that production technology followed Cobb–Douglas production functions. Equation (2) can be written as follows:

$$Y_{it} = AK_{it}^{\alpha}L_{it}^{\beta}\theta_{it} \tag{3}$$

Transforming (3) into a linear expression by taking logarithm both side of the equation, we have:

$$\ln Y_{it} = \ln A + \alpha \ln K_{it} + \beta \ln L_{it} + \ln \theta_{it} \tag{4}$$

Assuming $\theta_{it} = e^{u_{it}}$, we can rewrite (4) as:

$$\ln Y_{it} = \ln A + \alpha \ln K_{it} + \beta \ln L_{it} + u_{it} \tag{5}$$

From Equation (5), the natural logarithm of the TFP index is equal to the residual term $u_{it}$ in the econometric production function.

In order to examine the firm TFP and its determinants, this research used panel data procedures since the sample contained firm and overtime data. Therefore, Equation (5) can be estimated using common panel data estimation techniques, including fixed effects (FE) and random effects (RE) models. The FE regression took into account the individual effects of each firm included in the samples, thus it allowed the variation in intercept for each firm while the slope coefficients are constant across firms. Another technique was the RE model, which is based on the assumption that the group or individual effects can be estimated and they are not correlated with other independent variables. In this study, both FE and RE techniques to estimate the production functions in Equation (5) were employed.

### 2.2. Measuring Determinants of Firm Productivity

In the second stage, with the firm productivity measure having been estimated, the factors that explain a significant proportion of the variability in productivity of the Vietnamese agricultural sector were identified. We considered a comprehensive set of firm-relevant variables proposed in various strands, but they have not been generally combined into a single analytical framework. To specify, the role of each of the following factors: firm size, firm age, status of ownership, exporting, internet usage and access to finance in determining firm TFP was investigated (Table 1).

**Table 1.** The definition and summary statistics of variables.

| Variables | Description | Number of Observations | Mean | SD |
|---|---|---|---|---|
| LnY | The logarithm of value added (Y) calculated by subtracting intermediate, indirect, and raw materials costs from the total revenue from sales at the end of the survey year. | 4200 | 5.2391 | 1.8769 |
| LnK | The logarithm of capital (K), which is defined as the total book value of assets at the end of the survey year | 4200 | 7.9674 | 2.0649 |
| LnL | The logarithm of labor (L) is defined as the total number of regular workers at the end of the survey year. | 4200 | 3.6411 | 1.4854 |
| TFP | Total Factor Productivity (in logs) by FE | 4200 | 5.3863 | 1.5467 |
| | Total Factor Productivity (in logs) by RE | 4200 | 5.2513 | 1.3580 |
| VERY SMALL | Dummy variable = 1 if the number of employees is less than 10. | 4200 | 0.1183 | 0.3230 |
| SMALL | Dummy variable = 1 if the number of employees is equal or large than 10 and smaller than 200. | 4200 | 0.7464 | 0.4351 |
| LARGE | Dummy variable = 1 if the number of employees is large than 300. | 4200 | 0.1057 | 0.3075 |
| AGE | Number of years since the firm established (in logs) | 4200 | 2.3408 | 0.6421 |
| STA | State ownership, as percentage of state ownership of the firm | 4200 | 0.3074 | 0.4573 |
| FOR | Foreign ownership, as percentage of foreign ownership of the firm | 4200 | 0.0381 | 0.1847 |
| EXP | Export (=1 if the firm exports their products) | 4200 | 0.0452 | 0.2078 |
| ITN | Dummy for internet access (=1 if the firm has internet access) | 4200 | 0.1821 | 0.3960 |
| LOAN | Dummy variable for bank loan (=1 if the firm reports that it has a bank loan) | 4200 | 0.7302 | 0.4438 |

SD: Standard Deviation.

To examine the determinants of firm productivity, we first estimated TFP as residual $u_{it}$ from Equation (5). The variables Y, K, and L are derived from the survey data as follows:

- Value added (Y) is calculated by subtracting intermediate, indirect, and raw materials costs from the total revenue from sales at the end of the survey year.
- Capital (K) is defined as the total book value of assets at the end of the survey year.
- Labor (L) is defined as the total number of regular workers at the end of the survey year.

Then, the Equation (6) was estimated, of which the obtained TFP is a dependent variable whereas explanatory variables are set for firm-specific factors:

$$u_{it} = \sum_j \gamma_j F_{ijt} \qquad (6)$$

where:

$F_{ijt}$: firm-specific factors $j$ of firm $i$ at time $t$

$\varepsilon_{it}$: "White noise" error terms

It is worthwhile to note that the estimation of Equations (5) and (6) may suffer from several potential econometric problems. Accordingly, the empirical results should be interpreted with caution. First, the possibility is that endogeneity issues and reverse causality for several variables may arise in these two equations (Fernandes 2008). In order to address these problems, several control variables including firm size and year fixed effects were added. These variables control potential unobserved factors that may affect the TFP and its determinants. As a consequence, the estimation of TFP determinants was less subject to the endogeneity issues. Second, since the study employed a set of different variables to explain the TFP, it is likely that two or more variables in the regression correlated with each other, or, in other words, a multicollinearity problem may arise. Our approach to deal with this problem is to check the robustness of the main models by estimating regressions with a single independent variable at time along with basic control variables (firm size and year fixed effects).

*2.3. Data*

To examine the firm productivity and its determinants, we used firm-level data from the Annual Survey on Enterprises collected by the GSO (General Statistical Office, Vietnam), covering a 10-year period from 2000 to 2009. The data was considered as the most comprehensive data set available on Vietnamese firms. It contained information about the type of enterprise, ownership status, number of employees, assets and liabilities, sales, capital stock and obligations to the government. The survey covered all state owned and foreign owned firms without any threshold of firm size. The enterprises were categorized into four groups according to their number of employees at the end of the surveyed years: (i) very small sized firm (less than 10 workers); (ii) small sized firm (10 to 200 workers); (iii) medium firm (200 to 300 workers); and large-sized firms (more than 300 workers).

In undertaking the analysis of the data, there were several challenges occurred. Some variables of the input data, consisting of firms' year of establishment and export status were not available every year for all firms. In addition, there were a number of reentry enterprises that disappeared and reappeared later. The information on firms that either merged or changed their main business activities was also not covered by the survey. Therefore, it was necessary to take several steps to clean up the data. In this study, a balanced panel data set used for regressions was constructed. Thus, only firms with information on inputs, output and cost shares available every year during the research period were uniquely used. In addition, reentry enterprises were omitted from the data since the information on these firms for the missing years could not be collected.

Each firm in the survey was provided with a unique special tax code that remained unchanged over the years, which allowed us to generate a panel data set following individual firms. In addition, the survey contained information on the firms' industry codes, defined by the 2-digit VSIC 1993 (Vietnamese Standard Industrial Classification 1993). Therefore, firms that operate in the agricultural sector during the research period were filtered. To specify, the sample of the study included firms that belong to one of the following groups: VSIC 01- "Agricultural, hunting, and related service activities"; VSIC 02- "Forestry and related activities" and VSIC 05- "Fishing, operation of fish hatcheries and fish farms; service activities incidental to fishing". The final data was a balanced panel dataset of 420 enterprises mobile in the agricultural sector from 2000 to 2009. It is also important to note that the agricultural firms in this study differed from household farms and businesses. The formers are

registered as enterprises with the government and are governed directly by the Enterprise Law while the latter are not officially registered and are laid down in a government decree.

*2.4. Statistical Analyses*

The hypothesis that each coefficient is different from 0 was tested by the *p*-value. To reject this, the *p*-value had to be lower than a certain value. There were three different threshold levels including 0.1, 0.05 and 0.01 were considered, if a smaller *p*-value indicated the variable had more significant influence on the dependent variable.

## 3. Results

*3.1. Descriptive Statistics*

The definition and summary statistics of variables used in the study were described in Table 1. The TFP (in logs) was examined by fixed effects (FE) and random effects (RE) models. Although there were four sizes of enterprises categorized; very small, small, medium, and large (<10, <200, <300, and >300 employees, respectively)—only three firm sizes can be placed in the inputs of the model to compare with the omitted medium size < 200 < 300 employees. It was found that, in the agricultural sector in Vietnam, firms at small size (< 10 < 200 employees) accounted for 74.6%, whereas the numbers of companies at very small and large sizes distributed 11.8 and 10.6%, respectively. Accordingly, the medium firms < 200 < 300 employees displayed 3.0% of the total agricultural enterprises in Vietnam.

Consequently, state ownership explained 30.7% of the total agricultural enterprises, whilst the foreign ownership accounted for a much lesser number (3.8%). In addition, exporting firms allocated 4.5% (Table 1). There was a high proportion of firms that carried a bank loan (73.0%), but in contrast, only 18.2% of agricultural companies had access to the Internet (Table 1).

*3.2. Measuring Determinants of Firm Productivity*

Table 2 shows the determinants of TFP obtained from the two models including fixed effects (FE) and random effects (RE). They consist of firm sizes and age, ownership (state and foreign), export, and accessibility to Internet and bank loan. The two models FE and RE were controlled for the year fixed effects (Table 2). To recognize whether the individual effects were fixed or random, the Hausman specification test, of which the null hypothesis preferred RE against FE models, was employed. The *p*-value from the Hausman test suggested that FE regression was more effective than the RE models. Hence, the specification of TFP by FE as the benchmark estimation was obtained whilst the results of the RE model were reported as supplementary information.

Results in Table 2 showed that both very small and small firms exerted significant but negative TFP impacts (−71.3% and −32.1%, respectively; $p < 0.01$), whilst the large size enterprises exhibited a remarkable and positive TFP value (38.8%; $p < 0.01$), as compared to the medium size enterprises, which were omitted from regressions. During 2000–2009, firms >200 workers were more productive than the very small and small size (<200 employees), of which enterprises >300 workers obtained the most dynamic TFP values (Table 2).

The analyses also revealed that the age of firms displayed a negligible and statistically insignificant TFP value (3.2%). The state ownership factor accounted for 30.7%, but it possessed a trivial and unfavorable TFP (−7.5%). In contrast, the foreign ownership contributed only 3.8%, but it exhibited a maximal impact on TFP (55.0%; $p < 0.01$), as compared to the other variables (Table 2). Subsequently, the agricultural exporters accounted for 4.5% (Table 1), and they did not result in a significant TFP (2.6%; Table 2). Another important result was that enterprises with a bank loan (73.0%; Table 1) caused notably positive effects on TFP (18.5%; $p < 0.05$; Table 2).

**Table 2.** Total factor productivity of agricultural firms and its determinants.

| Variables | Dependent Variables of TFP | |
|---|---|---|
| | Fixed Effects (FE) | Random Effects (RE) |
| **VERY SMALL** | −0.7133 *** (0.0427) | −0.8559 *** (0.0381) |
| **SMALL** | −0.3208 *** (0.0388) | −0.4633 *** (0.0345) |
| **LARGE** | 0.3877 *** (0.0418) | 0.5214 *** (0.0372) |
| **AGE** | 0.0316 (0.0239) | 0.1000 *** (0.0202) |
| **STA** | −0.0752 (0.0767) | 0.9775 *** (0.0438) |
| **FOR** | 0.5496 *** (0.1641) | 1.0656 *** (0.0970) |
| **EXP** | 0.0255 (0.0255) | 0.0422 * (0.0246) |
| **ITN** | 0.0341 ** (0.0176) | 0.0505 *** (0.0157) |
| **LOAN** | 0.1847 *** (0.0121) | 0.1603 *** (0.0111) |
| Constant | 6.2873 *** (0.0815) | 5.7344 *** (0.0702) |
| Year fixed effects | Yes | Yes |
| Observations | 4200 | 4200 |
| Hausman test | | 473.32 *** |
| F-test | 1117.88 *** | 2562.44 *** |
| R-squared | 0.8429 | 0.8077 |

Values reported in parentheses are robust standard errors (SE) *, ** and *** indicated significance at 10%, 5%, and 1% levels, respectively. The Hausman test indicated a specification test of which the null hypothesis preferred a model that had random effects versus fixed effects.

## 4. Discussion

This study was the first to evaluate the TFP of agricultural firms in Vietnam during 2000–2009 from multiple variables corresponding to firm size and age, ownership, accessibility to loan and Internet, and export activity. Other researches on TFP in the agricultural sector in Vietnam principally employed a single variable framework (Doan et al. 2014; Huang and Zhang 2016; Tran 2014; Pham 2015). For instance, Doan et al. (2016) studied the misallocation and productivity in Vietnamese manufacturers and found that the large companies faced more constraints than the small enterprises. The findings of our research were reversed, as it was shown that the agricultural firms in Vietnam were 86.4% <200 employees, but less productive than firms >200 employees (Table 2). Few studies have employed the multivariate framework. Tran and Keishiro (2012) conducted a survey of 200 agro-enterprises in North Vietnam and revealed that the number of year in land using, Internet accessibility, and regional advantages exhibited positive impacts on TFP of agro-enterprises in Vietnam. Ho (2014) applied the Tornqvist model to evaluate the TFP of agriculture at local levels across 60 provinces in Vietnam during 1990–2006. There were four groups of determinants including omitted inputs of the agricultural production process, quality of inputs in agricultural production, technology factors, and output structure. It was reported that the TFP varied among regions in Vietnam, of which the

South provinces were more productive than the others. Land quality and fragmentation and farm size showed significant impacts on TFP (Ho 2014).

With respect to the impact of firm size and age, our empirical results demonstrated that firms >200 workers were more productive than the very small and small sizes (<200 employees), of which enterprises >300 workers were the most productive ones. These findings were in line with those of Van Biesebroeck (2005) for African economies and Satpathy et al. (2017) for India but were in contrast to those of Srithanpon (2016) and Fernandes (2008) for the case of Thailand and Bangladesh, respectively. Van Biesebroeck (2005) also noted that larger firms were shown to be more productive in developed countries like the United States. Turning to the effects of firm age, it has been theoretically suggested that due to learning by doing, firm productivity would increase with age. Our results also showed that older firms tend to have higher TFP than the younger ones (3.2%, Table 2). However, the impacts were not statistically significant in our benchmark FE model. This result might be attributed to the fact in the Vietnamese agricultural sector that some old firms still relied on traditional techniques in production while young ones tend to apply new technological advancements, which enabled them to reduce the experience gaps with older competitors and improve their productivity.

Another factor that possibly affects the firm TFP is the firm's status of ownership. Two variables (STA and FOR) representing the share of state and foreign ownership, respectively, were included in our regression. With regard to the effects of state ownership, it is argued in the literature that state-owned enterprises had some advantages in terms of governmental policies such as licensing, tax breaks, low interest loans and grants of land, resulting in higher productivity than private-owned ones (Li et al. 2014). However, in this study, we could not find strong evidence to support the link between state ownership and firm TFP. The coefficient of this variable was negative and statistically insignificant. This can be explained by the fact that, in a transition economy like Vietnam, the operation of some state-owned enterprises is inefficient due to their backward technology, soft budget constraints and inflexibility in adjusting transition process. Regarding the impacts of foreign ownership, foreign-owned firms are expected to have higher TFP than domestic ones since the former tends to have a higher level of investment and technology usage (Newman et al. 2009). Furthermore, in developing countries, foreign-owned firms are likely to have advantages in terms of both tangible assets, such as technology, and intangible assets, for example better access to distribution and marketing channels and networks (Arnold and Javorcik 2005). Our findings were in line with the above studies since it indicated a notably positive effect of foreign ownership on firm TFP. However, it is important to note that the foreign ownership accounted for a relatively small fraction of total agricultural firms (3.8%, Table 1). Furthermore, foreign direct investment into Vietnamese agricultural sector remained limited as it was shown that only 1.1% of foreign funds pouring into the country were cultivating the agricultural sector (FIA 2017). Therefore, it is crucial for the government to attract more foreign investors in this sector by introducing preferential policies such as reducing import-export tariffs, cutting corporate income tax, and cutting rental fees.

Regarding the impact of exporting, the advantage of TFP of exporters may be due to the learning mechanism in which enterprises learn from international customers and obtain experience from international competition. On the other hand, exporters were enabled to improve their own technological capabilities to exploit profitable opportunities in export markets (Fernandes 2008). Thus, a positive relationship between exports and firm TFP is expected. However, in this study, significant impacts of exporting on firm TFP were not achieved. This may be due to the fact that Vietnamese agricultural exporters had to face a number of challenges in the international market, including complex and rigid trade barriers and fierce competition. Many Vietnamese agricultural products challenged difficulties in developed markets, which have gradually grown to saturation and protectionism. As a consequence, the exporters were not always more productive than the non-exporters, as observed from the TFP value (Fernandes 2008).

The accessibility to the Internet was a productivity-enhancing factor in the agricultural sector. There were multiple ways in which the Internet can help to improve firm productivity in developing

countries (Paunov and Rollo 2015). It enabled firms to access relevant market information for smaller and informal businesses. Furthermore, the Internet may facilitate more effective firms' production and delivery chains, and create new business opportunities (Kaushik and Singh 2004; Aker and Mbiti 2010). In this study, we included a dummy variable for Internet access in our regression to investigate the role of the Internet in determining firm TFP. Our results revealed that firms with access to the Internet were 3.4% more productive than those without Internet. This finding was in line with results reported by Tran and Keishiro (2012) and the World Bank (2016b), which indicated an increase of 1.9% in TFP growth for firms using the Internet in Vietnam.

Turning to the effects of access to finance, we used a dummy variable for having a bank loan (LOAN) to investigate the impact of this factor on firm TFP. We found that firms with access to a bank loan were 18.5% more productive than those without. Our findings were similar to those obtained by Dollar et al. (2005) who found a positive relationship between access to finance and firm TFP for the cases of Bangladesh, India, China and Pakistan. However, this study did not analyze how effective agricultural enterprises were at using the bank loan for their production activities. Further data should be collected in the period 2010–2019 to clarify this point in order to establish the countermeasures to the next 10-year period 2020–2029.

Comparing the agricultural TFP in developing countries, Coelli and Rao (2005) conducted a survey in the agricultural sector of 93 developed and developing countries that accounted for principal world population and agricultural output during 1980–2000. It was found that the TFP change of Asia was the maximum (1.029), followed by North America (1.027), whilst Africa displayed the lowest one (1.013). Of which, Vietnam was ranked 17th among the 93 countries worldwide, similar to Sudan and Bangladesh (1.024). China and Cambodia were highest (1.060 and 1.057, respectively), whereas Haiti and Chad were lowest (0.957 and 0.947, respectively) (Coelli and Rao 2005). Seker and Saliola (2018) recently examined the TFP of 69 developing countries of their manufacturing firms, with the determinants including exporting, innovation, access to finance, foreign ownership, and regulations. They used six different models to estimate the TFP and all countries were ranked with respect to these specifications. The TFP of Vietnam in 2009 was in third place, in the same group with Botswana 2006, Cote d'Ivoire 2009, Uzbekistan 2008, Mexico 2010, and Syria 2009, while first place included Brazil 2009, Jordan 2006, Kyrgyzstan 2009, Cameroon 2009, Ethiopia 2006, and Indonesia 2009. In Southeast Asia, the TFP of Vietnam stood behind Indonesia 2009, but it was greater than Malaysia 2007 (4th place) and Thailand (7th place) (Seker and Saliola 2018). Comparing with other continents, Asian countries including Vietnam have shown a positive growth of TFP, but in contrast, the growth of TFP in the agricultural sector of 30 African countries was −0.15% annually (Avila and Evenson 2010).

It was also found that the factors that affect TFP varied among regions of a country, differed amidst nations and continents, and between developed and developing countries. For instance, in South Asia, the major drivers of agricultural TFP growth were the levels of natural, human and technology capital endowments. In contrast, financial capital and crop diversification had opposite effects on TFP (Anik et al. 2017). On the other hand, the TFP growth in Czech depended rather on natural and weather conditions (Macheck and Spicka 2013). Helfand et al. (2015) analyzed the TFP in Brazil's agricultural firms and found that it grew annually by 5% during 1985–2006, but turned to slow growth of TFP by 1.74%. The reasons that caused the reduced TFP value were the inadequate investment of public infrastructure and technical assistance from extension service to agricultural enterprises. For instance, in 2006, only 22% of Brazilian farms received technical assistance, and recently it became a policy priority of Brazil to increase the TFP in the agricultural sector (Helfand et al. 2015). The references of how to improve the TFP from neighboring countries in Southeast Asia, South Asia together with lessons from decreased TFP in Brazil may help Vietnam to successfully boost the productivity of agricultural firms and hence devote the sustainable growth of the country.

## 5. Conclusions

This study investigated the determinants of firm productivity by exploiting a dataset of 420 Vietnamese agricultural enterprises from 2000 to 2009. Productivity was measured as TFP obtained by a production function estimation using fixed and random effects models. The relationship between TFP and a comprehensive set of firm-specific variables including firm size, firm age, status of ownership, exporting, internet usage and access to finance in determining firm TFP was examined. The empirical results showed that firm size, foreign ownership, Internet, and credit accessibility were positively correlated with firm TFP. The findings of this study were effective for policy makers since it suggested where the reform efforts should be placed. Since foreign owned firms were found to be the most productive, attracting foreign direct investment in agriculture may provide more benefits for firm productivity in this sector. Reform efforts should also focus on improving the productivity of small enterprises, specifically firms <200 employees. Furthermore, it is crucial for the government to provide a good regulatory framework for specific services that agribusiness firms require, including infrastructure and access to financial services. The utilization of achievements from this research may help promote the TFP in the agricultural sector in Vietnam and thus effectively contribute to the stability and sustainable growth of the country.

Although our findings yield important policy implications and contribute to the determinants of firm productivity analysis literature, there are still some limitations to this research. First, the estimation of the effects of some variables on firm TFP may lead to problems of causality. For example, with the access to finance variable, it is probably true that more productive firms tend to make greater use of credit than less productive ones, as it is more profitable to the former. To solve this causality problem, it is important to find instrumental variables for explaining variation in the access to finance of agricultural enterprises in Vietnam. Therefore, more empirical efforts are needed to enable the discharge of spurious effects, as well as to establish real causal relationships between the firm-specific factors and productivity. Second, using a fixed effects estimation for the production function can partly address the endogeneity bias by eliminating unobserved fixed firm characteristics that may simultaneously affect input choices and TFP. However, the possibility is that there may still be unobserved time-varying firm characteristics that simultaneously affect input choices and TFP. Hence, alternative approaches should be considered to address this problem.

Beside those limitations that must be overcome, the research also can be further developed in some other directions. First, it is essential to compare the productivity and its determinants of firms in this study with those of household farms and businesses who play an important role in agricultural productions in Vietnam. This can be done by exploiting the annual Vietnamese household surveys conducted by GSO. Second, further analysis on how effective agricultural firms used bank loans to increase higher TFP, as well as continuous analyses on other TFP determinants during the period 2010–2019 should be carried out to establish the necessary countermeasure for the duration of 2020–2029. Because Vietnam is of industrializing and urbanizing, improving the TFP in agricultural production is indispensable for the stability and sustainability of the country, at least by 2030. Third, although Vietnam has witnessed impressive improvement with respect to irrigation facilities in recent decades, the agricultural firms have been still in the face of intensifying competition for land, water, and budgetary resources (World Bank 2016a). These might be important factors that directly and indirectly affect firm productivity. Studies on the impact of these factors can provide insights for policy-makers and agricultural firms for a more efficient and sustainable use of land and water.

**Author Contributions:** T.D.X. designed the overall research, conducted the research findings, and revised the manuscript. M.H.G. is the main writer of the paper, collected data and analyzed the results. M.T.Q. and B.H.T. contributed to the model analysis. All authors revised and approved the final manuscript.

**Funding:** This research received no external funding.

**Acknowledgments:** Yuichiro Yoshida is appreciated for assistance in analytical methods of this research.

**Conflicts of Interest:** The authors declare no conflict of interest.

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
