# Peer review of "Total Factor Productivity of Agricultural Firms in Vietnam and Its Relevant Determinants"

_economies, doi:10.3390/economies7010004_

Round 1

Reviewer 1 Report

I agree with the conclusions of the article (which is technically sound) that large agribusiness firms, if private, are more likely to be productive. This has little to do with the 40% of workers vs. <20% of GDP in agriculture they use to start the article. Most farmers are in family farms, not in agribusiness firms.

More generally, I suspect that the type of crop (or animal) has a lot to do with productivity in firms. Directions for future economic research would be if coffee, rubber, or other plantation crops can get the kind of productivity gains that fish or shrimp ponds can. There is no doubt that better clonal varieties and careful fertilization, weeding and irrigation will increase output per hectare for coffee and rubber, but less evidence on labor productivity. If TFP data by crop were available (and I am not suggesting this article has to address the crop issue), it would help identify policies including crop/animal R&D.

But the macro-problem is that 40% of workers are in a low-return industry with limited growth prospects. In the last decade, 95-100% of population growth has been in urban areas, so younger workers are “solving” the problem gradually. Changes to land ownership (make land ownership or control more secure for farmers) would help allow for naturally larger farms with more mechanization, as elsewhere. Then more capital would cooperate with limited labor and produce higher incomes. This is being impeded by land policy – probably because local governments use land takings at a low price and resale at a higher price after rezoning it as a revenue source – that hurts the rural sector.

As to what to do with the article: I would publish it if the distinction between agricultural firms and family farms were made clear and directions for future research were suggested. These are minor edits, not major changes. A quibble is if things like Internet access cause productivity or are correlated with other things that “really” cause TFP, but this is minor and endemic for this kind of analysis.

The other caveat that might be inserted is that many agribusiness firms are using up groundwater and are not sustainable. Water and soil mining do not show up in TFP but could end up creating large flows of workers into urban areas, if the eco-systems collapse.

Author Response

Dear Respective Reviewer

We would like to reply you your comments and questions with details are as below. Thank you very much.

Question: I suspect that the type of crop (or animal) has a lot to do with productivity in firms. Directions for future economic research would be if coffee, rubber, or other plantation crops can get the kind of productivity gains that fish or shrimp ponds can. There is no doubt that better clonal varieties and careful fertilization, weeding and irrigation will increase output per hectare for coffee and rubber, but less evidence on labor productivity. If TFP data by crop were available (and I am not suggesting this article has to address the crop issue), it would help identify policies including crop/animal R&D.

Answer: Thank you for your valuable comments. The type of crop or animal was an important factor driving the productivity of agricultural firms. Although the information on specific types of crop and animal was not available for this dataset but firms in the survey were categorized into three groups including Growing of crops and farming of animals, Forestry and Fishing. Thus, we added dummy variables to the model to examine whether the TFP differed among these sub-sectors.  The results showed that firm in Forestry and Fishing sectors were 43.3% (p<0.05) and 68.5% (p<0.05), respectively, more productive than those involved in growing crops and farming of animals activities. In other words, firms operating in Fishing sector exhibited the most remarkable TFP during the research period. We added these analyses in the Abstract, Results, Discussion and Conclusion parts. As edits were marked by red letters, please kindly check.

Question: As to what to do with the article: I would publish it if the distinction between agricultural firms and family farms were made clear and directions for future research were suggested. These are minor edits, not major changes.

Answer: We have added the distinction between agricultural firms and household businesses in the Data part, Line 205-209. The formers are registered as enterprises with the government and are governed directly by the Enterprise Law while the latters are not officially registered and are laid down in a government decree. In Vietnam, The Enterprise Law of 2005 specified that “household businesses frequently using 10 employees or more have to register as an enterprise governed by this Law". However, the fact is that there have been still a large number of household businesses were reluctant to switch their business model since it would lead to more costs and disruption in their current businesses. In future researches, it is essential to compare the TFP and its determinants of firms and household businesses since it may yield important policy implications. This was discussed in the future research directions of the study, Line 416-420. Please kindly check.

Question: A quibble is if things like Internet access cause productivity or are correlated with other things that “really” cause TFP, but this is minor and endemic for this kind of analysis.

Answer: Thank you for this comment. The estimation of the effects of some variables on firm TFP may lead to problems of causality. For example, with the access to finance variable, it is probably true that more productive firms tend to make greater use of credit than less productive ones, as it is more profitable to the former. To solve this causality problem, it is important to find instrumental variables for explaining variation in the access to finance of agricultural enterprises in Vietnam. Therefore, more empirical efforts are needed to enable the discharge of spurious effects, as well as to establish real causal relationships between the firm-specific factors and productivity. We discussed this issue in Line 403-411. Please kindly check.

Question: The other caveat that might be inserted is that many agribusiness firms are using up groundwater and are not sustainable. Water and soil mining do not show up in TFP but could end up creating large flows of workers into urban areas, if the eco-systems collapse.

Answer: Thank you for this comment. Since the dataset did not cover information on the use of groundwater, we were not able to examine the effects of this factor on firm TFP. However, this suggests a new research direction in future if data on land and water use of agricultural firms is available. This can be done in future by designing a comprehensive survey that includes all relevant factors that potentially affect the firm productivity. This issue was discussed in Line 423-428. Please kindly check.

Reviewer 2 Report

This paper is poorly written. It was very hard to read. In addition, the data used for the study is from 2000 to 2009, and does not provide adequate insights to readers of today (2018/2019), for there are so much that has changed over the years.

Author Response

Dear Respective Reviewer

We would like to reply you your comments and questions with details are as below. Thank you very much.

Question: This paper is poorly written. It was very hard to read.

Answer:  Thank you very much for your valuable comment. We have carefully revised our manuscript following your suggestions with all parts having been rewritten in a more readable way. We removed many redundancies, discussed more about motivations and explained in detail the reasons why we conducted this research in the Introduction part. The Discussion part was also rearranged more appropriately and included some detail discussions about the impacts of independent variables on the TFP. In conclusion part, main findings and contributions of the study were summarized. Furthermore, we added some discussions on the limitations of the study and future research directions.  We marked the revisions by red color letters throughout these parts. Please kindly check.

Question: In addition, the data used for the study is from 2000 to 2009, and does not provide adequate insights to readers of today (2018/2019), for there are so much that has changed over the years.

Answer:  Thank you for this comment. We have discussed about this issue in the Discussion and Conclusion part as future research direction. One of the main purposes of our research was to provide empirical evidences on the determinants of firm TFP during the period 2000-2009 and hence, suggested a framework to assess the TFP and potential factors affecting TFP in future research. It is important to conduct further analyses on TFP determinants in the period 2010-2019, especially how effective agricultural firms used bank loan, should be carried out to establish necessary countermeasure for the duration 2020-2029. Furthermore, there were a number of studies on the firm productivity, which have recently published, based on dataset collected from last decade. These studies not only provided framework for future research but also yield important policy implications. To illustrate, for the case of Vietnam, Ho (2014) applied the Tornqvist model to evaluate the TFP of agriculture at local levels across 60 provinces during 1990-2006. Based on World Bank's survey data in 2004 and 2005, Trung and Kaizoji (2017) investigated the impact of investment climate on firm productivity. In other countries, Bournakis and Mallick (2018) reviewed the state of the art in firm-level TFP by employing panel of firms in the UK during 2004-2011. Another study of Dai, Li and Lu (2017) explored how urbanization economies impact the TFP of research and development performers using a sample of Chinese enterprises during 2005-2007. Please kindly check the following references:

1. Ho, D. Bao. 2014. Provincial Total Factor Productivity in Vietnamese Agriculture and Its Determinant. Journal of Economics and Development 6(2): 5–20.

2. Trung, N. Ba, and T. Kaizoji. 2017. Investment Climate and Firm Productivity: An Application to Vietnamese Manufacturing Firms. Applied Economics 49(44): 4394–4409.

3. Bournakis, I., and S. Mallick. 2018. TFP Estimation at Firm Level: The Fiscal Aspect of Productivity Convergence in the UK. Economic Modelling 70: 579590.

4. Dai, M., X. Li, and Y. Lu. 2017. How Urbanization Economies Impact TFP of R&D Performers: Evidence from China. Sustainability 9(10), 1766.